# Adult Skeletal Age-at-Death Estimation through Deep Random Neural Networks: A New Method and Its Computational Analysis

**DOI:** 10.3390/biology11040532

**Published:** 2022-03-30

**Authors:** David Navega, Ernesto Costa, Eugénia Cunha

**Affiliations:** 1Centre for Functional Ecology (CEF), Laboratory of Forensic Anthropology, Department of Life Sciences, University of Coimbra, 3000-456 Coimbra, Portugal; eugenia.m.cunha@inmlcf.mj.pt; 2National Institute of Legal Medicine and Forensic Sciences, 3000-548 Coimbra, Portugal; 3Centre for Informatics and Systems of the University of Coimbra (CISUC), Evolutionary and Complex Systems Group (ECOS), Department of Informatics Engineering, University of Coimbra, 3030-290 Coimbra, Portugal; ernesto@dei.uc.pt

**Keywords:** forensic anthropology, age-at-death estimation, machine learning, neural networks

## Abstract

**Simple Summary:**

Age-at-death is of paramount importance in forensic analysis of skeletal remains. In addition to sex, stature, and population affinity, it constitutes baseline information in the identification process of deceased individuals. Despite its long tradition, in anthropological research age-at-death estimation poses many challenges and unanswered questions. It is undisputedly among the most difficult tasks of the forensic anthropologist and its results are often subject to a lackluster performance. In this study, we assessed computationally the efficiency of a holistic approach to skeletal age estimation based on a new proposal for macroscopic examination and the use of machine learning-based models for data analysis. Our results suggest that this approach is key for accurate and efficient age-at-death estimation based on skeletal remains analysis.

**Abstract:**

Age-at-death assessment is a crucial step in the identification process of skeletal human remains. Nonetheless, in adult individuals this task is particularly difficult to achieve with reasonable accuracy due to high variability in the senescence processes. To improve the accuracy of age-at-estimation, in this work we propose a new method based on a multifactorial macroscopic analysis and deep random neural network models. A sample of 500 identified skeletons was used to establish a reference dataset (age-at-death: 19–101 years old, 250 males and 250 females). A total of 64 skeletal traits are covered in the proposed macroscopic technique. Age-at-death estimation is tackled from a function approximation perspective and a regression approach is used to infer both point and prediction interval estimates. Based on cross-validation and computational experiments, our results demonstrate that age estimation from skeletal remains can be accurately (~6 years mean absolute error) inferred across the entire adult age span and informative estimates and prediction intervals can be obtained for the elderly population. A novel software tool, DRNNAGE, was made available to the community.

## 1. Introduction

Forensic anthropology (FA) has become a major component of forensic sciences. During recent decades, a profound change, a true paradigm change, has taken place and forensic anthropology has transformed itself into a discipline with its own theoretical and conceptual corpus and research agenda. It can be stated that the discipline and its attributes have evolved significantly. In fact, this evolution has been so marked and drastic that it can be argued that even some of the most experienced and long-term practicing anthropologists may have trouble conceptualizing and being fully proficient in the many areas now covered by the discipline [1,2], or in even being able to foresee all possible interdisciplinary and technological developments. Nonetheless, biological profile estimation from human skeletal remains constitutes a pivotal task and inferring age-at-death, sex, stature, and population affinities is a fundamental step of the anthropological analysis in the context of the medico-legal identification process.

In the identification process of human remains, age-at-death is a major screening factor that helps reduce the universe of possible matches. Therefore, an estimate of this biological parameter is a normal request from police forces and judicial entities [3]. This process relies on a meticulous analysis of skeletal and dental structures with an association with chronological age-at-death. Although this is a topic in which significant research has been performed in recent decades, skeletal age estimation of adult remains continues to present many unanswered questions and challenges, especially for the elderly. Determining how to handle age estimation using multiple skeletal age-related traits remains among the problems most commonly identified for which a satisfactory solution has not yet been presented and research further is required [3,4,5,6,7,8,9,10]. Moreover, computational and statistical methods employed in the creation of age estimation techniques have been a topic of debate and contention [11,12,13,14,15,16,17,18,19,20,21,22,23,24].

The present work aims to lay a foundation to tackle some of the challenges of morphoscopic adult skeletal age estimation, especially in terms of its holistic or multifactorial aspect. Several authors argue in favor of multifactorial age estimation to obtain precise and accurate age estimates [9,16,25]. Nonetheless, multifactorial age estimation poses its own challenges and limitations, and is a topic with a clear lack of consensus [5,10]. Conceptually multifactorial age estimation can be argued as being the most effective approach for age estimation because morphological indicators display different age-related trajectories and have different underlying biological processes.

The symphyseal face of the pubic bone, for instance, has been systematically studied, ranging from the pioneering studies that established the morphological analysis of this skeletal marker as an age estimation technique, to modern fully computational frameworks for age estimation [26,27,28,29,30,31,32,33,34]. However, other skeletal markers and regions that can convey important age-related information, such as the degeneration of vertebral bodies, joint margins, or the roughening of muscle and tendon attachment sites, have received scarce attention as aging markers. The unimpressive accuracy and precision associated with the multiple iterations of pubic symphysis aging techniques, one of the most used and favored techniques for age estimation [5], underlines the idea that further developments and over-analysis of specific skeletal markers in isolation is not likely to result in substantial improvements over the state-of-art of adult age estimation, but rather a more comprehensive array of skeletal markers and features provide a more fertile ground for further developments [35,36].

A multifactorial morphoscopic approach to skeletal analysis does not solve, in itself, the many difficulties faced in the age-at-death assessment. In fact, if not correctly designed, this approach can become methodologically cumbersome from a data collection and analysis perspective. From an analytical and statistical perspective, collecting more data from the skeleton increases the chance of encountering issues of redundancy, multicollinearity, and a dimensionality that hinders the straightforward interpretability and pragmatic value of morphoscopic analysis. From a practical point of view, a more comprehensive analysis of the age-related skeletal features requires a higher level of expertise on how to collect the skeletal features. This issue is of great relevance for approaches that rely on morphoscopic analysis of the skeleton. Moreover, in forensic contexts it is common that the skeletal remains are somehow fragmentary or incomplete due to a multitude of taphonomic factors, which means that not all age-related traits will be available for every unidentified deceased. From a practitioner’s perspective, this translates into the need for computational and software tools that can fit or train age-at-death estimation models on a case-by-case basis.

To cope with the difficulties and needs of multifactorial age estimation, novel methods and techniques can be developed by resorting to statistical and machine learning, data science, and artificial intelligence tools and approaches. More than constantly evolving, machine learning, artificial intelligence and data science are ubiquitous, and have various successful applications within forensic anthropology in domains such as biological profiling or craniofacial identification [13,15,37,38,39,40,41].

This work aims to provide a new method, and its computational analysis, for multifactorial skeletal age-at-death estimation of adult humans supported by a machine learning approach based on a deep randomized neural network. This manuscript is in its essence methodological, presenting both a new macroscopic technique for skeletal analysis and a detailed explanation of a computational framework to obtain age-at-death estimates and model their uncertainty. New age-at-death estimation software, DRNNAGE, that translates the in silico key points of the work presented here into an actionable tool, was developed and is a major research product.

## 2. Materials and Methods

### 2.1. Dataset

#### 2.1.1. Sampled Identified Skeletal Collections

To implement and pursue a computational analysis of the novel age-at-death estimation method proposed in this work, a reference dataset of 500 individuals was constructed. A total of 99 features were collected covering all key traditional age-related and other under-explored skeletal traits. Accounting for laterality, 64 unique traits can be analyzed from the axial and appendicular skeleton using the new macroscopic scoring method, whose rationale and details are described and explored in Section 2.2.

The 500 individuals were sampled from two identified skeletal collections hosted at the Department of Life Sciences at the University of Coimbra, Portugal—the Coimbra Identified Skeletal Collection (CISC) and the 21st Century Identified Skeletal Collection (XXI-ISC). The CISC consists of 505 individuals with age-at-death ranging from 7 to 96 years representing skeletons from the Cemitério da Conchada, that were born between 1817 and 1924 and died from 1904 to 1938 [42]. The XXI-ISC collection is currently composed of 302 skeletons of both sexes, mostly represented by elderly individuals. This collection represents Portuguese nationals who died between 1982 and 2012 and were exhumed between 1999 and 2016 from a main cemetery in Santarém. More details are found in [43,44]. Demographic parameters of the sampled individuals in our study are detailed in Table 1. All sampled individuals presented fully developed long bones. No individual was excluded due to pathology or taphonomy.

The sampled reference dataset is composed of 250 male and 250 female individuals who died at the age of 19 to 101 years old (mean = 57.34, SD = 22.93). Age-at-death distribution is homogenous across the age span represented, with the exception of individuals over 95 years old (Figure 1). A homogenous and uniform age-at-death distribution is a simple yet vital strategy to cope with the problem of age-mimicry [45] and to guarantee that the targeted age span is fully represented. 

Sampled individuals were born between 1830 and 1982 and died between 1910 and 2012. Despite the large temporal frame represented, there is a continuum and a wide range over the age-at-death distribution that makes this sample particularly suited for age-related research.

#### 2.1.2. Data Management and Processing

As previously mentioned, multifactorial age estimation poses many challenges that are mostly related to data management and processing. Two common problems that arise are redundancy and missing data. Redundancy is always involved when bilateral or paired data is collected. The human body is not fully symmetric; yet it is not expected that the left and right diverge drastically under normal conditions. Missing data in FA results mostly from taphonomic factors. To cope with redundancy and missing values, a strategy based on domain heuristics and imputation techniques was pursued. For bilateral traits, the left side was selected as the main source of data. If the left score for a given bilateral trait was missing, the right side was used as a surrogate value. Once this first heuristic was applied, the remaining missing values were imputed using a simple nearest neighbor (k = 1) procedure by substituting all missing value of given individual by the values of the nearest neighbor. Jaccard similarity on one-hot encoded data was used to compute the nearest matches. The followed procedure minimized redundancy and dimensionality by reducing the number of skeletal features from 99 to 64. A simple nearest neighbor with k = 1 according to Beretta and Santianello [46] is the preferred strategy to preserve the structure of a dataset. The authors demonstrated that more advanced algorithms reduced imputation error but introduced significant data distortion. To increase the volume and age-related variability of the data available, sexes were pooled. Although this choice seems arbitrary, it is important to note that, in FA, sex is usually estimated during casework. Pooled data models balance out the potential and pitfalls of sex-specific models and their mis-specifications.

Missing values represented 9.52% of the total entries of the data table when bilateral data were considered, and 6.89% when the domain heuristic described was first applied as a naïve imputation mechanism and strategy to handle bilateral data redundancy.

### 2.2. A Novel Technique for Macroscopic Age-At-Death Estimation

A key contribution of the present work to the topic of macroscopic skeletal age estimation in adults is the proposal of new scoring schemes for well-established and underexplored skeletal traits that can be used as biomarkers in age-at-death assessment. The development of a new scoring system emerged from the necessity for standardization of a data collection, and a generation mechanism that was more aligned with a multifactorial approach to age estimation and more suitable multivariate data analysis, while keeping in mind practical aspects such as observation error and ease of application.

The proposed morphoscopic method strives to be comprehensive and to incorporate features from as many skeletal elements as possible. Envisioning the whole skeleton as a biomarker for age estimation, it is more likely that the overall skeletal patterns exhibit a stronger and monotonic relationship with age-at-death, which is pivotal for accurate predictions. The rate and nature of overall skeletal changes also have a greater chance to be consistent across individuals since a holistic approach can encapsulate intra and interpersonal variation with greater finesse [35]. Analyzing multiple traits also offsets the intrinsic limitation to specific traits when analyzed on their own [47].

Following a component-based approach, up to 64 unique skeletal traits can be scored using the scheme outlined in the next subsections. The covered skeletal traits encode both developmental and degenerative aspects from different anatomical regions. Despite the large number of features analyzed in this proposal, all skeletal features are limited to morphological variables with no more than three classes or stages. Such specifications were established during the several iterations of the development and refinement of the system proposed, and by following guidelines from the literature. Shirley and Montes [48] empirically addressed the old methodological debate of phase versus component-based approach. Their study quantified the observation error of a phase and a component-based method, and the results suggests that a component-based approach offers a more objective scoring if the number of coding possibilities in each component does not exceed three levels of expression.

The following subsections provide a brief overview of the existing scoring methods for specific skeletal region or traits, the novel scoring schemes proposed in this work, and the rationale and difficulties faced during method development. Due to the constraints of space and manuscript presentation, full descriptions of the trait scoring systems developed in this study are provided in Appendix A. The skeletal scoring systems are also embedded in the developed software (see Section 2.6.4).

#### 2.2.1. Cranial and Palatine Suture Scoring

The scoring system used for the cranial and palatine sutures consists of a modification and binarization of the proposal by Boldsen et al. [19]. This system was selected because it incorporates much of the rationale of older methods for scoring ectocranial sutures (neurocranium) and the palatine sutures [49,50,51,52,53,54,55,56]. The simplification to a binary scoring system resulted from the difficulty during preliminary and training sessions to differentiate and consistently score the adjacent stage (i.e., open to juxtaposed or partially obliterated to punctuated). The scoring scheme described in Appendix A should be applied to nine sutural segments from the palatine, the sagittal, coronal, and lambdoid sutures (Appendix A).

#### 2.2.2. Vertebrae Development and Degeneration Scoring

The fusion of the bodies of the first and second sacral vertebrae is also part of the skeletal markers analyzed in the proposed protocol. This skeletal feature is one of the few developmental traits that persist through early adulthood. Its usefulness as an indicator to distinguish young adults was demonstrated by several researchers [57,58,59]. This trait was assessed with a binary scale described in Appendix A. To incorporate both metamorphic and degenerative traits of the vertebral column, a three-stage scoring scheme was devised, building upon previous work from Snodgrass [60], Watanabe and Terazawa [61], and Albert et al. [62]. The first two methods focus on the degeneration and osteophyte formation on the margins of the vertebral bodies, whereas the last work focuses on the development of the vertebral epiphyseal rings and body morphology. The proposed system, Appendix A, applies to superior and inferior surfaces of the third to seven cervical vertebrae, the first to fifth lumbar vertebrae, and the superior surface of the first sacral vertebra. Appendix A lists all features analyzed in the axial skeleton (excluding sacral auricular surfaces).

#### 2.2.3. Joint and Musculoskeletal Degeneration Scoring

Osteoarthrosis and entheseal changes have been traditionally analyzed in physical anthropology and bioarcheology as markers of health and biomechanical stress, and tentative indicators of physical activity patterns. According to Milner and Boldsen [35], who advocate a more detailed analysis of this type of skeletal marker, these features collectively contribute to an increase in accuracy and precision of age estimation. The authors base such an assertion on empirical evidence from an experience-based procedure where these types of skeletal traits were extensively used. Several reasons can be noted for why osteoarthrosis and entheseal changes have been overlooked or not systematically analyzed in the past as age markers. Broadly speaking, due to their degenerative nature and late onset, it is believed that they provide limited information, distinguishing only in a broad sense young from older individuals. More specifically, osteoarthrosis increases with age but has a complex and multifactorial etiology that hinders or masks its relationship with age-at-death. Entheseal changes have traditionally been assessed as musculoskeletal stress markers and as tentative clues to infer physical and occupational activity. This possible relation to activity can interfere in the expression and variation of entheseal morphology and affects its relationship with the aging process. However, recent and systematic studies conducted on identified skeletal collections show that age-at-death is one of the most relevant factors, or even the only one with statistical significance, in the expression of such skeletal traits [63,64,65,66,67,68,69,70].

Developing a scoring procedure for these features proved to be one of the most challenging aspects of method development. The difficulties faced were mostly related to the fact that analyzing joint and musculoskeletal degeneration involves many skeletal elements, which translate into high dimensionality of the collected data. This high dimensionality poses two major problems: increased chance of collinearity, which poses computational issues, and loss of pragmatic value. To tackle the high dimensionality and subsequent issues found when scoring joint and musculoskeletal degeneration, a new binary procedure was developed. The system retains the analysis of the type of traits evaluated in Buikstra and Ubelaker [71] and Henderson et al. [72] but simplifies the scoring to a simple absence or presence of degenerative traits as a whole for any particular anatomical structure. The generic binary scoring system both for joint and musculoskeletal degenerative changes are presented in Appendix A. The scoring system applies to five major anatomical complexes from the upper and lower limb: shoulder, elbow, hip, knee, and ankle (Appendix A). To enhance the analysis of these traits we provide specific scoring descriptions for Stage 1 of some traits (Appendix A).

#### 2.2.4. Clavicle Sternal and Acromial Ends Scoring

The macroscopic analysis of the clavicle has a long standing in skeletal age estimation. Nonetheless, its focus has been mostly in the epiphyseal fusion of the sternal end [73,74,75,76]. Sternal epiphyseal fusion of the clavicle is a key trait to obtain precise age estimate in young adult individuals due to the late total development of this structure around the 30 s. Falys and Prangle [73] were the first to propose a method to score post-epiphyseal changes in the clavicle for age estimation purposes. The authors suggest a scoring system focused on surface topography, porosity, and marginal osteophyte formation, providing a regression model for age estimation. A new scoring scheme that integrates both developmental and degenerative changes in the sternal and acromial ends of the clavicle is proposed. A full description of the traits analyzed is available in Appendix A.

#### 2.2.5. First Rib Costal Face and Tubercle Scoring

The metamorphosis of the sternal end of the ribs emerged in the mid-1980s as a new age estimation technique. İşcan, Loth and colleagues described multiple morphologic features that characterize the metamorphosis of the sternal end of the ribs, with particular emphasis on the fourth rib costal face [77,78,79,80]. This approach proved to be an effective alternative to existing methods. Nonetheless, several disadvantages have been pointed out, such as the difficulty in identifying the fourth rib in disarticulated skeletal remains and the fact the morphology of the costal face is not the only component of the age-related changes in rib morphology. To address these problems, Kunos et al. [81] described a new age estimation method based on the metamorphosis of the costal face, head, and tubercle of the first rib. The first rib has the key advantage of having a morphology that is straightforward to individualize. DiGangi et al. [82] improved upon the work of Kunos et al. [81] and proposed a revised method for age estimation based on the costal face and tubercle morphology. A new scoring method is proposed in this study that build upon previous work by Kunos et al. and DiGangi et al. [81,82]. This new system simplifies the scoring of the costal face morphology to a three-stage coding and the morphology of the tubercle is evaluated in a binary fashion (Appendix A).

#### 2.2.6. Pubic Symphysis Scoring

The metamorphosis of pubic symphysis is the most popular osteological marker used in adult skeletal age estimation. The previous attention paid to this anatomical structure is not misplaced; however, the over-reliance on this indicator can be explained by the progressive metamorphic features that have enough expression variation to allow an exhaustive morphological description using different scoring schemes and different types of supporting materials such as casts. A simple component-based system was developed focused on the metamorphic and degenerative changes in three features of this structure: rim development, topography, and texture of the symphyseal face. These three components are assessed with a three-stage coding system emphasizing early metamorphic or development traits, such as the presence of billowing (a pattern of transverse ridges and furrows) and late degenerative traits, such as the flattening and erosion of the symphyseal face. A full description of the scoring system is given in Appendix A. The proposed system is based on previous work by Todd [30,31] and Brooks and Suchey [26].

#### 2.2.7. Sacral and Iliac Auricular Surfaces (Sacroiliac Joint) Scoring

The description of age-related changes in the sacro-iliac joint can be traced back to Sashin [83] and Schunke [84], but its usage as an age indicator its mostly due to the work of Lovejoy and colleagues [85] and Buckberry and Chamberlain [86] on the chronological metamorphosis of the iliac auricular surface, and the age estimation method by Passalacqua [59] based on metamorphic and degenerative changes in the sacrum.

To incorporate age-related features of the sacro-iliac joint, a two-component-based system was developed to assess textural and marginal changes in the sacral and iliac auricular surface. The iliac and sacral auricular surfaces undergo textural changes that are characterized by the transition from a smooth, finely grained surface to a granular, irregular and porotic surface. The margins that delimit the surface tend to manifest osteophytic activity as age progresses. Both the texture and margin features refer to the entire structure but very often the degenerative changes, in particular the margin, are more pronounced in specific areas such as the inferior and anterior apexes. Full features descriptions are given in Appendix A.

#### 2.2.8. Acetabulum Scoring

Several age-related changes can be documented in the acetabulum and used for age estimation [87,88,89,90,91,92,93,94]. One key aspect of the acetabulum is the late onset of the age-related changes and its durability and resistance to taphonomic factors. To incorporate this skeletal element in our protocol, a three-stage scoring system for the changes occurring on the rim, posterior horn, and acetabular fossa was developed. In the spirit of Calce [90], who simplified the method developed by Rissech et al. [91,92], the foundation of the scoring system presented in Appendix A is based on a simplification and adaptation of the method proposed by San-Millán et al. [87,95].

#### 2.2.9. Scoring Reliability: Intra-Observer Error

To assess the reproducibility of this new proposed scoring system, 50 individuals were randomly selected and rescored on all possible traits (m = 99) by the first author. For bilateral traits, only the left side was used for further intra-observer reliability analysis (first author) to avoid issues that arise from non-independent ratings. Kendall’s W [96] was computed as a concordance coefficient to assess consistency between scoring sessions. This metric ranges from 0 (no agreement) to 1 (perfect agreement).

### 2.3. Feature Analysis Via Sphering and Marginal Correlation Analysis

To assess the relationship of the analyzed traits with age-at-death, we inspected marginal correlation coefficients using Spearman’s correlation coefficient (ρ) and Pearson’s eta coefficient (*η*^2^). In addition to these two coefficients, we also computed marginal correlations adjusted for inter-trait correlation following Zuber and Strimmer [97]. This technique aims to cope with the myopy of univariate feature selection methods by computing marginal correlations of decorrelated predictors with the target class. First, the data centered and scaled, and then transformed by applying a linear basis that enforces orthogonality among predictors while maintaining the maximum relationship with the original standardized predictors. After this transformation, also known as the Mahalanobis transform or sphering, the predictors covariance matrix is the identity matrix (no correlation). The authors called the adjusted marginal correlations CAR scores and proved that ranking based on these quantities provides a fast and optimal procedure for feature ranking and selection. We suggest [97,98] as primers on feature selection and data sphering based on this approach.

### 2.4. Randomized Neural Networks: Theory and Implementation

From a computational perspective, age-at-death estimation can be viewed as a function approximation problem, $y=f^{*}\left(x\right)$, and constitutes one of the core reasons why artificial neural networks were chosen as the predictive technique in this work. In age-at-death estimation, $y=f^{*}\left(x\right)$ maps the input skeletal traits (*x*) to an age-at-death (*y*). ANNs are function approximation machines that define the mapping y=f(x;θ), where *θ* are the parameters or network weights that result in the best approximation [99].

Artificial neural networks are a class of connectionist, biologically inspired computational models that enable learning from data for a multitude of tasks, such as classification, regression, representation learning, and data compression and generation. ANNs are, in a broad sense the result of two components: architectural design—that is how many layers and neurons comprise the network; and an optimization algorithm—how the parameters of the network are learnt.

In its basic implementation, an ANN is composed of three layers: an input layer, a hidden layer, and an output layer. Two sets of weights are embedded in the network structure: one connecting the inputs to the hidden layer and the other connecting the hidden layer to the output layer. In a neural network, the input is transferred to the hidden layer by means of a non-linear activation function. An activation function and the set of weights define a node of the hidden layer. Such nodes are also known as artificial neurons. An artificial neuron, the key component of an ANN, is a mathematical operator in the form of:(1)$$h(x)=g(\sum_{i=1}^{p}x_{i}\omega_{i}+b)$$
where g() is an activation or transfer function, $x_{i}$ and $\omega_{i}$ are the *i*-th components of the input, and the weight vector *b* is the neuron bias. Artificial neurons are, in essence, non-linear functions with learnable parameters, which ultimately expand the ANN model representational capacity to be able to approximate any output function.

A key aspect of ANN is their flexibility and modularity, which due to their capability can be applied to a vast array of heterogeneous data types and domains. The explosion in the availability and capacity to store and analyze data in the form of images, video, audio, and unstructured text has led to the development of novel ANN training algorithms and architectures, and a transition from shallow (single hidden layer) to deep (multi-layer) networks. It is important to note that not all ANNs are formulated and trained in the same manner. There are specialized architectures to tackle; for instance, data in the form of images that make use of computational operations, such as convolutions and pooling. However, a transversal aspect of modern ANNs is their use of gradient-based learning algorithms, where the parameters of a network are iteratively fine-tuned. Gradient-based learning enables end-to-end training and state-of-the-art performance in many complex tasks, but it is costly and requires considerable amounts of technical knowledge to leverage an ANN to its full potential.

A counterintuitive, yet highly efficient, approach to the training of ANN models is to randomly assign and fix a subset of parameters (i.e., hidden weights) of the network and recast the optimization component to a simpler least squares estimation problem [100,101]. In the context of ANNs, randomization as an intrinsic mechanism of model learning can be traced back to late 1980s and early 1990s, with the proposal of randomized radial basis functions network (RBF) and the random vector functional link network (RVFL) models [102,103,104,105,106]. However, the recent interest in randomized algorithms for training feed-forward neural networks can be attributed to the re-emergence of this approach in the guise of the controversial extreme learning machine (ELM) algorithm [107,108,109,110]. According to [111], there is no need to rename this strategy for training neural networks, since all key elements have been previously proposed [102,103,104,105,106], and some of the minor changes introduced by the ELM algorithm, such as the omission of direct links between the input and output layer—present in the RVFL network—can have a deleterious effect in performance. Nonetheless, the ELM algorithm acted as a foundation for many innovations in the field of randomized artificial neural networks (RANNs), such as the development of highly efficient algorithms to compute and cross-validate the output layer analytically [112,113], and its evolution from a framework restricted to shallow networks to a set of techniques and algorithms capable of deep, multi-layered network architectures [114,115,116,117,118].

#### 2.4.1. Efficient Training and Regularization in Randomized Neural Networks

In randomized neural networks, the elements of $\omega_{i}$, the hidden layer weights, are randomly generated from a suitable probability distribution and are not optimized. Only the output weights are learned from data by solving a least squares estimation (LSE) problem expressed as:(2)$$\beta =H^{†}Y$$
where *β* are the output layer weights, $H^{†}$ is the Moore–Penrose pseudo-inverse of the matrix *H*, which defines the hidden layer, and *Y* is a column vector storing the network target output, in our case, age-at-death. $H^{†}$ can be computed using several methods; a common approach is through orthogonal projection using Equation (3):(3)$$H^{†}={\left(H^{T}H\right)}^{-1}H^{T}$$

From Equations (2) and (3), it is trivial to show that the use of this algorithm yields an age estimate as $\hat{Y}=H\beta$, and that the output layer is in fact an ordinary least squares linear regression built on the non-linear feature mapping induced by the hidden layer of the neural network.

It has been noted [119] that one can keep the algorithmic simplicity of the least squares solution, while improving its performance and generalization capability by adding a penalty to the output weights. Such a penalty, *C*, stabilizes the inversion of matrix *H* and shrinks the coefficients of the output layer towards zero; smaller coefficients lead to smaller error rates on unseen data. Imposing such a constraint on the output weights is a process known as shrinkage or regularization, which in the neural network literature is also named weight decay. This type of regularization is also referred as L2-norm regularization or Tikhonov regularization.

The solution of a regularized RANN is obtained by fitting a ridge regression model [120] as the output layer. The ridge solution, β_ridge_, is obtained by substituting Equation (3) as follows:(4)$$H^{†}={\left(H^{T}H+\frac{I}{C}\right)}^{-1}H^{T}$$

I refers to the identity matrix with dimensions matching $H^{T}H$. Regularization is of paramount importance when training a randomized neural network for age estimation. The solution of the network is obtained by minimizing the squared error as the objective function. LSE-based neural networks lead to unbiased solutions but with high variance if not properly regularized due to the randomness of the initialization [112]. Regularization shrinks the size of the output coefficients towards zero, which is consistent with the theory that smaller weights result in better generalization of neural networks [121,122].

Since the output layer in a RANN is solved as a least squares estimation problem, fortunately, there exist highly efficient, analytical, and closed formulations to assess the leave-one-out (LOO) error, as shown by Shao and Er [112] using Allen’s [123] Prediction Sum of Squares (PRESS) statistic:(5)$$E_{LOO}=\frac{1}{n}\sum_{i=1}^{n}{\left(\frac{y_{i}-{\hat{y}}_{i}}{1-h_{ii}}\right)}^{2}$$
where *h_ii_* is the *i*-th diagonal element of the hat or projection matrix, which is the matrix that maps the hidden layer parameters to the predicted values of the network, in our case age-at-death. Shao and Er [112] have demonstrated that computing the projection matrix of the network and finding the optimal regularization parameter, *C*, under leave-one-out cross-validation (LOO-CV), can be achieved with computational efficiency by performing a singular value decomposition (SVD) of the hidden layer, which, given such an operation, is written as $H=U\Sigma V^{T}$. Using SVD, the network estimate can be written as:(6)$$\begin{array}{c} \hat{Y}=H\beta \\ \hat{Y}=H{\left(H^{T}H+\frac{I}{C}\right)}^{-1}H^{T}Y\\ \hat{Y}=U{\left(\Sigma^{T}\Sigma +\frac{I}{C}\right)}^{-1}\Sigma^{T}U^{T}Y \end{array}$$
where $U{\left(\Sigma^{T}\Sigma +\frac{I}{C}\right)}^{-1}\Sigma^{T}U^{T}$ is the projection matrix and it can be noted that only ${\left(\Sigma^{T}\Sigma +\frac{I}{C}\right)}^{-1}\Sigma^{T}$ affects the projection matrix for different values of *C*. Σ is a diagonal matrix whose element are expressed as $\phi_{i}=\frac{\sigma_{ii}^{2}}{\sigma_{ii}^{2}+\frac{1}{C}}$, where $\sigma_{ii}$ is the *i*-th singular value from the decomposition of H. SVD makes the regularization of the neural network highly efficient because the diagonal of the projection matrix, which is needed to calculate the LOO error using Equation (6), can be obtained from the following Hadamard products (matrix element-wise multiplication):(7)$$\gamma =U\circ \Gamma^{T}=U\circ (\Theta \circ U^{T})$$
where $\Theta ={\left(\Sigma^{T}\Sigma +\frac{I}{C}\right)}^{-1}\Sigma^{T}$. The diagonal elements of the projection matrix, *h_ii_*, can be obtained by performing a column-wise sum of the elements of γ. The LOO predictions of the network can be obtained analytically as follows:(8)$${\hat{y}}_{i}=\frac{y_{i}-f(x_{i})}{1-hat_{ii}}$$

In addition to this highly efficient computational strategy to train a randomized neural network, data standardization and the addition of Gaussian noise to several of the components of the network can also improve performance and accuracy.

#### 2.4.2. From Shallow to Deep Randomized Neural Networks 

The mathematical and network formulation presented above pertain to a randomized weights single layer network architecture. Navega and Cunha [124] introduced this model in skeletal age estimation in the formulation of the ELM network (no direct links in the network) and applied it to several traits of the sacroiliac joint. However, several authors proposed different techniques to extend the RANN to deeper architectures [114,115,116,117,118]. To increase the deepness of the network, one can resort to fully randomized approaches or use autoencoding strategies and stack multiple autoencoding RANNs to build a multi-layer network. In this work, due to its simplicity, we follow the proposal of Shi et al. [118] to train deep randomized network models (DRNNs). Following the authors, the first layer of the network is defined as:(9)$$H^{\left(1\right)}=g(XW^{\left(1\right)})$$
where *X* is the input matrix, in our case skeletal traits. Every subsequent layer (*j* > 1) is defined as:(10)$$H^{\left(j\right)}=g(H^{\left(j-1\right)}W^{\left(j\right)})$$
where *H*^(*j*−1)^ is the previous layer. One can also allow connections from the input to all hidden layers and define the hidden layer as:(11)$$H^{\left(j\right)}=g([H^{\left(j-1\right)}\;X]W^{\left(j\right)})$$
where $W^{1}$ and $W^{j}$ are the weight matrices between the input-first hidden layer and the inter-hidden layers, respectively. These matrices are randomly assigned and held fixed during the training. The input to output layer is then defined as:(12)$$D=\left[H^{\left(1\right)}\;H^{\left(2\right)}\;\ldots \;H^{\left(j-1\right)}\;H^{\left(j\right)}\;X\right]$$

The design of the deep network is very similar to that of a shallow RANN, and it can be easily seen that the input to output layer consists of non-linear features induced by the hidden layers concatenated to the original input of the network. When the input is reused directly in the output layer, the network is classified as a network with direct link or skip layers. As mentioned above, this is the key difference between ELM and RFVL networks.

#### 2.4.3. Deep Random Neural Networks as Implicit Ensemble Models

One key advantage of the randomized approach used in this study is that it can enable implicit neural ensemble models [118]. Rather than applying Equation (2) once to solve the output layer weights (solution), Equation (2) can be re-used along the depth of the network for each $H^{\left(j\right)}$ computed from Equations (9) or (10), and obtain an intermediate age-at-death estimate. The final age-at-death estimate can be then obtained by averaging all estimates along the network depth. This feature stabilizes the predictions and offers a different mechanism to train an ensemble model other than training each model independently.

### 2.5. Regression Uncertainty Modeling and Prediction Intervals

The approach followed in this work relies heavily on regression. In Section 2.4.1 and Section 2.4.2, we presented the foundation for mathematical age-at-death prediction using RANN models as a regression task. However, we focused only on how point estimates can be obtained, that is, the conditional expectation of age-at-death given a specific skeletal pattern of an individual. Mapping the uncertainty of the point estimate is essential in forensic anthropology, which means that a predictive interval for a preset confidence level should also be part of the analysis and the subsequent report.

In the current work, we follow a simple and generic approach based on modeling the conditional variance associated with each point estimate (network prediction). We recast the prediction interval construction as a regression problem and, using LOO network predictions, we build a regression uncertainty model (RUM) by regressing absolute residuals on predicted age-at-death. We then scale the predicted residual by 1.2533 to obtain a standard deviation associated with each age estimate. The scaling factor is the ratio of the standard deviation to the absolute deviation [125,126]. Assuming normality of the variance around each point estimate, the prediction interval associated with an ANN model is given by the quantiles of a Gaussian or truncated Gaussian parameterized with the conditional mean and standard deviation inferred from the ANN and its associated RUM. The key advantage of this approach is its simplicity compared to likelihood methods [15,16,17,20,23,127,128,129] or conformal prediction theory, as in [113,124,130]. In addition to the numerical interval, this approach also allows visualization, as illustrated by Figure 2.

### 2.6. Computational Analysis: Design, Parameterization, Metrics, and Software

#### 2.6.1. Experimental Design

To assess the performance of DRNN and Gaussian RUM models in multifactorial age estimation from macroscopic skeletal traits we followed a simple template for robust metric assessment based on a resampling Monte Carlo cross-validation (MCCV) scheme. This works as follows: for a given iteration of the scheme, split the dataset into disjoint train and test partitions. Using the training partition, fit a DRNN and RUM models by making use of Equations (5)–(7) to optimize the regularization parameter *C* and obtain leave-one-out predictions. *C* is optimized as $2^{x}$ with x∈{−6,−4,…,12}. With the trained DRNN and RUM models, we predict the age-at-death of the testing sample/partition and compute the MCCV performance metrics. For a given set of skeletal traits, this procedure is repeated 1000 times (B = 1000). The training partition is set as 80% of the total data (400 of 500) and the test partition as the remaining (100 of 500). This sampling procedure was performed without replacement. The core of our computational analysis is organized in two experiments, from now on referred to as experiments A and B:(A)The first experiment we conducted was designed to provide a baseline of the accuracy obtained by fitting DRNN models to blocks of traits that have standard or traditional analytical framing. For instance, we fitted models to different anatomical complexes or sets of traits that mimic existing aging standards, i.e., a model for the sutures or the pubis symphysis.(B)Our second computational experiment consisted of simulated different proportions of available traits from 90% to 10%. The objective of this experiment was to assess model performance in a more realistic scenario where the forensic anthropologist has skeletal traits available on a case-by-case basis.

In both experiments we computed 95% predictive intervals (95% PI) by setting the uncertainty of parameter σ = 0.05.

#### 2.6.2. Network Parameterization

A key aspect of any ANN model is its architecture, that is, how many neurons (or nodes) and layers comprise the network. To leverage the full potential of the DRNN, and to maximize its training speed and efficiency, rather than search for the optimal architecture, we developed a simple heuristic based on the work of Lappas [131]. The author demonstrated that the size of a single layer perceptron can be estimated from the number of samples available. Using his work as a foundation, we propose the following heuristics for setting the architecture of a DRNN. The width, size, or number of neurons of each layer was set as:(13)$$S=2^{\left\lfloor \log_{2}\left(8\left(\sqrt{2^{k}}/k\right)\right)\right\rfloor },\;\;k=\log_{2}(n)$$
where *n* is the number of samples. The depth or number of layers was set as:(14)$$L=2^{\left\lfloor \log_{2}(k)\right\rfloor },\;\;k=\log_{2}(n)$$

Following Equations (13) and (14) as a simple heuristic allows us to have predictable, parsimonious network architectures. In this way, the network allows many computing units for randomized feature extraction distributed over several layers without incurring overparameterization. This heuristic also leverages the simplicity of training a deep neural network using the same mechanisms of a shallow one, while exploiting an implicit ensemble framework (Section 2.4.3). For our experiments, applying the described heuristic defines the network architecture with a rectangular topology comprising eight layers of 32 neurons each, for a total of 256 randomized units.

DRNNs are computationally cheap nonlinear models built by combining regularized linear regression with nonlinear features obtained by using an activation function, g(.), with random weights. In this work, we used the rectified linear unit (ReLU) as the nonlinearity of the networks. The ReLU is defined as g(z,w)=max(0,zw), where *z* and *w* are the layer input and random weight matrices. Since the regularization process involved in the training process described in this work is not scale invariant, during network training normalization by mean centering and variance scaling, Equation (6) was performed on the matrices *X*, *XW*, *H*, and *Y*. The output of the network was later rescaled before computation of the performance metrics.

ANN architecture selection and design is a non-trivial task often performed through very expensive and complex computational strategies and procedures. The heuristic used and architecture selected in this work emerged from trial-and-error experimentation during the development of the *rwnnet* software package (see Section 2.6.4). This parameterization leverages the benefits and key features of randomized neural networks—fast training and prediction with minimum technical knowledge, given that the model is fully described through linear algebra and matrix operations.

#### 2.6.3. Performance Metrics

In our analysis, we evaluate four parameters that any model used in regression task should have, especially one used for age estimation. An age-at-death prediction model—regardless of its underlying mathematical algorithm—should be accurate, unbiased, valid, and efficient. Accuracy refers to the ability of the model of the model to predict age with minimal error. The most straightforward metric to assess this parameter is the mean absolute error (*MAE*) computed as:(15)$$MAE=\frac{\sum_{i=1}^{n}\left|y_{i}-{\hat{y}}_{i}\right|}{n}$$
where $y_{i}$ and ${\hat{y}}_{i}$ are the known and predicted values, respectively, and *n* is the number of evaluated samples.

A model should be unbiased, that is, free of systematic error. A typical pattern of bias or systematic error in age estimation models is the over-estimation of young individuals and under-estimation of the elderly. A robust and comprehensive way to assess bias (${\hat{\beta }}_{e}$) is by computing the slope of the regression line of the residuals, $e_{i}=y_{i}-{\hat{y}}_{y}$, on known values. When minimal to no bias is presented, this value should be close to zero. A positive slope suggests a systematic bias, such as the one describe previously. Bias is computed as:(16)$${\hat{\beta }}_{e}=\frac{\sum \left(y_{i}-\bar{y})(e_{i}-\bar{e}\right)}{\sum {\left(y_{i}-\bar{y}\right)}^{2}}$$
where $\bar{y}$ and $\bar{e}$ are the means of the known and residual values.

The validity of model, in the context of our study, refers to the ability of a model to contain the known age within the predictive interval and within a reasonable margin close to the nominal uncertainty level allowed. For instance, for an uncertainty level (alpha) of 0.05 (or 5%) we expect that the coverage of the correct proportion of individuals within the predictive interval is close to 0.95 (or 95%). As a validity measure, we compute:(17)$$P(\alpha )=\frac{\sum_{i=1}^{n}\delta (y_{i},l_{i},u_{i})}{n}$$
where $\delta (y_{i},l_{i},u_{i})$ is an indicator function with $\delta (y_{i},l_{i},u_{i})=1$, if $y_{i}\geq l_{i}\wedge y_{i}\leq u_{i}$ and $\delta (y_{i},l_{i},u_{i})=0$, and $l_{i}$ and $u_{i}$ are the values of the lower and upper ends of the predictive interval, respectively.

Finally, a model should thrive to be efficient. Efficiency in this context refers to the width or range of the prediction intervals associated with the regression uncertainty model. A method or model is efficient when it outputs the narrowest predictive interval possible while also maintaining its validity. We compute our measure of efficiency as follows:(18)PIW=Q(u−l,  τ),with τ ∈{0.5;  0.025; 0.975}
where *Q*(.) is a quantile function and τ a given quantile. One can see that we compute the median of the predictive interval width and its associated 95% confidence interval (quantile-base).

#### 2.6.4. Software

All computational work was performed using the R and C++ programming languages with all key software components written by the first author. To perform this work, the rwnnet, rumr, rmar, and lsmr packages were used. These packages are available from the respective repositories of the GitHub profile of the first author, https://github.com/dsnavega (accessed on 18 March 2022).

Novel software, DRNNAGE, that operationalizes age-at-death estimation following the macroscopic and computational techniques described in this work, was also developed and is live as a web application at https://osteomics.com/DRNNAGE (accessed on 18 March 2022).); its source is available at https://github.com/dsnavega/DRNNAGE (accessed on 18 March 2022).). In its current state, we strongly recommend that end users approach their analysis using only default parameters. All problems detected and suggestions should be directed to the corresponding author.

## 3. Results

### 3.1. Intra-Observer Scoring Error 

Overall, the new proposed macroscopic scoring technique presented high intra-observer consistency based on the results on Kendall’s W concordance coefficient [96]. With the exception of RD01 and FM01, 0.751 and 0.716, respectively, all skeletal traits presented a concordance coefficient higher than 0.800. The global average of this coefficient was 0.907. All traits presented a statistically significant concordance between scoring obtained by the first author in two different sessions. The high concordance observed can be explained by the simplicity of the scoring systems used with the large number of traits that were binary coded. Further inter- and intra-observer error analysis is required by an independent third party, due to the nature of the methods employed.

### 3.2. Marginal Correlation Analysis

Marginal correlation analysis showed that all traits have a statistically significant relationship with age-at-death. The cranial sutures showed the lowest marginal correlation (ρ: 0.297–0.519, *η*^2^: 0.088–0.249), with palatine sutures explaining less than 10% of the variation in observed age-at-death. The axial traits—cervical and lumbar vertebrae—exhibited a moderate to strong monotonic relationship and explained variation with age-at-death (ρ: 0.794–0.845, *η*^2^: 0.639–0.725). A similar correlation and explained variation pattern were observed for the clavicle traits (ρ: 0.710–0.851, *η*^2^: 0.507–0.729), first rib traits (ρ: 0.763–0.776, *η*^2^: 0.590–0.607), iliac auricular surface traits (ρ: 0.731–0.789, *η*^2^: 0.539–0.631), and the acetabular traits (ρ: 0.782–0.818, *η*^2^: 0.625–0.674). A slightly lower marginal correlation was observed for the pubic symphysis traits (ρ: 0.711–0.731, *η*^2^: 0.523–0.549) and sacral auricular surface traits (ρ: 0.632–0.704, *η*^2^: 0.398–0.499). Traits from the upper and lower limbs presented a wider range of correlation (ρ: 0.380–0.789, *η*^2^: 0.145–0.628). When analyzed in the context of feature ranking based on marginal correlations adjusted for inter-trait correlation (CAR scores), the suture traits score was among the worst predictors and its decorrelated components showed no statistically significant relationship with age-at-death. The several appendicular degenerative traits—HM04, UL01, RD01, FM01, FM02, and TB01—also showed no statistically significant correlation when assessed on a Mahalanobis transformed space. Ranking based on CAR scores showed that the top-ranking traits came from all anatomical regions rather than a specific indicator.

### 3.3. Computational Model Assessment

Results from the two in silico experiments performed to assess DRNN models in age-at-death estimation are reported in Table 2, Table 3, Table 4 and Table 5. Models based solely on the cranial sutures exhibited the worst performance among all models produced, having a median MAE of 15.300 (Table 2) and a median predictive interval width (PIW) of 68.144 years, which renders the cranial sutures an inaccurate and inefficient set of traits.

Modeling based on specific anatomical regions resulted in a DRNN with a median MAE ranging from 7.583 to 10.897 years (Table 2); focusing solely on this metric, it is reasonable to state that, on its own, different anatomical regions perform similarly in age estimation. The same can be said for the metrics of bias, validity, and efficiency. Predictive interval width is perhaps the most distinctive metric for practical applications. Anatomical regions with strong developmental signs, such as the clavicle or the pubis, tend to provide narrower predictive intervals for younger individuals.

Combining traits from different regions provided an improvement over models built on specific anatomic regions. Using 16 traits from standard age-related traits—clavicle, first rib, pubic symphysis, sacroiliac complex (auricular surfaces, S1 body surface, and S1-S2 fusion), resulted in a MAE of 6.609 (5.561–7.598, 95% CI) and reduced the prediction bias considerably when compared to any model built on the same anatomical regions independently (Table 2), and a PIW of 34.245 (12.927–41.087, PIW 95% CI). A model based only on degenerative traits (m = 39) resulted in a MAE of 6.962 (6.084–7.814, 95% CI) and median PIW of 33.732 (28.882–33.122, PIW 95% CI). From our results, multifactorial age estimation models provide improved efficiency, as reflected in narrower predictive intervals (Figure 3, Figure 4 and Figure 5).

From Figure 3, Figure 4 and Figure 5, we can also observe that multifactorial models provide accurate and efficient estimates across the entire adult lifespan, solving the problem of open-ended and unspecific age-at-death estimates for the elderly. Figure 4 illustrates the importance of non-standard traits to accurately predict advanced age-at-death. Based solely on degenerative traits of the vertebrae, limb joint, and musculoskeletal attachment sites, we can obtain estimates for the elderly that are comparable to more classical traits (Figure 3) or full-set models (Figure 5). The downside of relying solely on this type on indicator for age-at-death estimation is the wider intervals for young adults with no degenerative traits (95% PI ~18 to 46 years vs. ~18 to 32 if traits with sharp developmental stages are present).

The best performing models in experiment A were those built on the full feature set (m = 64), with a mean absolute error of 5.925 (5.110–6.728, 95% CI), and PIW of 30.010 (15.63–36.081, PIW 95% CI) years. The prediction bias for this model was 0.117 (0.060–0.170, 95% CI), which represents a two-to-six-fold reduction in the prediction bias compared to other models built on specific anatomical regions individually (Table 2). Results from experiment B (Table 4 and Table 5) showed that similar results can be obtained using different proportions of traits selected at random. 

An important remark to make regarding our results based on the two computational experiments is that analytical LOOCV, implicitly performed during model optimization, showed little to no disparity with the results obtained during the repeats of the Monte Carlo cross-validation procedure (B = 1000 repeats) where 20% of the data was used as a proper test set.

The accuracy of our approach can be visualized in Figure 6, where a scatter plot of known vs. predicted age-at-death is depicted. From this figure, one can infer that the predictions obtained using our approach maintain a similar level of error—dispersion around the identity line (dashed red line)—across the entire adult age span, and slightly more accurate for individuals under 40 years. For individuals over 90 years old at death, there is an observable under-estimation. It is also possible to visualize, Figure 7, that a deep RANN model using multiple traits produces minimally biased estimates.

Regarding the validity of the models trained in our computational experiments, results show that the predictive intervals contained the known age-at-death without significant deviation from the nominal level of uncertainty (median of P(α) ~ 0.95, with variation between 0.87 and 0.99). Multifactorial models also show a systematical reduction in prediction bias when compared to models based only on a specific anatomical structure.

## 4. Discussion

The main objective of this work was to investigate the fundamental issue of age-at-death estimation in the forensic analysis of human remains, and propose a new method and its computational analysis from a perspective of multifactorial analysis of the adult skeleton. Several age estimation methods have been previously developed, focusing on specific anatomical structures or regions such as the cranium, the ribs, or the pelvic joints. Nonetheless, it is well known that no single skeletal indicator is capable of producing accurate and efficient age estimates across the entire human age span. Determining how to report age estimates using multiple indicators or traits remains an open issue, with experts resorting to different heuristics that often are not standardized and lack a valid computational or statistical grounding [5]. In the literature, there are techniques that use multiple skeletal indicators for age estimation but are often limited to the cranial sutures and the pelvic joints [20,23,132]. More generic procedures for multifactorial analysis have also been proposed [133,134], but with poor adoption in forensic casework because they require seriation or advanced mathematical knowledge to be put into action. 

The current study provides strong support for multifactorial or multi-trait analysis of the skeleton as a way of obtaining accurate and efficient age estimates across the entire span of adulthood. Results from computational experiment A suggest that using each skeletal indicator or anatomical region separately provides limited improvement over existing methods. One striking remark from this experiment was the performance of the models solely based on the axial (vertebrae) and appendicular (limbs) skeleton. In previous studies, these traits have been considered to be only useful for providing a general estimate or limited in value for age prediction [135,136]; nonetheless, our results are consistent with those of more recent publications that assess their predictive utility and urge reconsideration of these traits as valid age-related traits [64,66]. For instance, if these traits all present a Stage 0, one can infer without any computation that the age-at-death of the deceased is between approximately 18 and 46 years (Figure 4, considering σ = 0.1). Our results also indicate that the inclusion of these traits is pivotal to solve the problem of open-ended age intervals and poor age estimation for the elderly. On their own, degenerative axial and appendicular traits allow estimation of the age-at-death of the elderly with an improved accuracy and efficiency compared to more standard traits such as the pelvic joints (i.e., pubic symphysis, acetabulum, iliac auricular surface). The neural model based on the full set of traits described in the novel macroscopic age estimation proposed here provided the best performance results in respect to all metrics analyzed. This can be attributed to the fact that having more features allows the deep neural models to operate at their maximum potential regarding what they do best—extracting novel features from existing ones using, in our case, random weights and a non-linearity (ReLU function) as a mechanism to combine multiple traits, which ultimately allows the output layer to operate in a non-linear regime, despite it being, in practice, a regularized linear model. Moreover, the multitude of traits scored also permits the models to encapsulate the intra- and inter-variability of skeletal morphology with greater finesse, which is manifested as more efficient (narrower) predictive intervals that reflect the heteroskedastic nature associated with the senescence process.

Although the main goal of the computational experiment A was to establish a baseline of performance of multifactorial age-at-death estimation compared to more traditional modeling approaches based on specific anatomical blocks or regions, experiment B aimed to assess the performance of neural models for age-at-death estimation in a more realistic setting, where the expert may not be able to use the pre-specified models or the full set of traits due to the availability of skeletal elements or the multitude of factors that make it impossible to score all traits defined in this macroscopic technique. This computational experiment also provides, both directly and indirectly, answers to several questions that may arise regarding the approach and technique used, and proposed in this work from a more pragmatical and casework view: Does the skeleton need to be complete to reap the maximum benefits of this protocol? Which combination of traits works best or is necessary? How practical is the method?

The results demonstrated that the accuracy of the full-set model (m = 64) can be maintained to large degree using smaller random combinations of traits, which ultimately are dictated on a case-by-case basis in a forensic setting. Once again, this can be explained by the capacity of the neural models to extract and combine information from the skeletal traits in an optimal way in terms of prediction. It is important to note here that models based on randomized proportions of traits presented performance metrics superior to most models based on specific anatomical regions, which reinforces our thesis that the multifactorial or multi-trait models are crucial for improving the state-of-art in forensic skeletal age estimation.

Finding an optimal or minimum number of traits is, from a combinatorial and practical point of view, an intractable problem, for which a solution can only be approximated with such a large number of traits (m = 64). However, such a solution would be computational wasteful and of little pragmatic value because, as in the situation of the full trait set, the optimal or minimum trait set can result in a non-applicable model due to the availability of skeletal elements during casework. This is the main reason why, in our study, we opted for a randomized evaluation of smaller traits set. Ultimately, we developed the DRNNAGE software to operationalize the age estimation procedure described in this manuscript, in a manner that is flexible and practical for the expert applying it, bearing in mind that each case will be limited by its own available skeletal traits. DRNNAGE allows the expert to compute the optimal network and associated uncertainty model based only on the traits that the forensic expert can score. Thus, in that regard, the usefulness of the estimates obtained is limited by biology and taphonomy itself, rather than the technical implementation.

From a practitioner perspective, marginal correlation analysis and the performance of the developed models clearly suggest that there is room for improvement in our approach regarding the issue of the traits to be used. For instance, our results suggest that there is little to be gained from including the cranial sutures, which, from a predictive modeling standpoint, resulted in the worst model on its own using our scoring protocol. Similar conclusions were reached by Jooste et al. [137], who also investigated the cranial sutures in the context of a multifactorial approach. To maximize the potential of the framework proposed in this work, it is important to bear in mind that domain and expert knowledge is of utmost importance; this can also be said of any other machine learning or computationally heavy approach. The practical aspect of this method can be improved if applied with the rationale of the well-known Two-Step Procedure proposed by Baccino et al. [138]. This procedure and heuristic for age-at-death estimation suggests age indicators should be combined logically or hierarchically rather than by brute force (i.e., averaging). In the context of our proposal, this translates into the following: if several traits with sharp metamorphic or developmental stages exhibit Stage 0—i.e., clavicle sternal end, S1-S2 fusion, pubic symphysis components—a neural model is trained using those traits and the other traits are ignored. The same rationale can be applied if the traits that encode a strong degenerative signal, such as the vertebrae and limb traits, are scored with their maximum stage (Stages 1 or 2). In this case, we have demonstrated that age estimation can be accurate and efficient when relying solely on these traits. As a final remark and suggestion to improve age estimation with our method, but also with any other method that employs a multifactorial or multi-trait approach, rather than focusing on an optimal or minimal number of traits to use, one should focus on the representational power of the traits analyzed and, whenever possible, use traits that represent both metamorphic and degenerative aspects of the skeletal development and senescence, as argued by Winburn [88].

The present work provides a solution to the problem of multifactorial age estimation based on the macroscopic analysis of the skeleton. Multifactorial skeletal age estimation is systematically noted as being the most accurate way to achieve an age estimation in adults, but is obtained through a plethora of procedures and heuristics that are often subjective and lack a clearly well-defined statistical or computational rationale [3,5]. As noted by Ritz-Timme et al. [3], a comparison of different methods with regard to their performance based on published data is an exercise that can only be undertaken with severe limitations and caution. The existing methods have been developed on samples of differing sizes, unbalanced age distributions, and different population backgrounds. There is no standardized array of statistical parameters used to assess an age estimation method, and different statistical procedures have been applied. In many cases, there is a lack of detail regarding the procedures used, and often only an incomplete analysis performance is pursued (i.e., focusing only on MAE and point estimate accuracy). In the context of our research subject, these limitations are exacerbated by the fact that, to the best of our knowledge, no other study in the literature has pursued a systematic analysis of adult skeletal age estimation using such a vast and diverse array of morphoscopic traits based on a single reference dataset. Nonetheless, a brief analysis of the most recent and comprehensive validation studies clearly demonstrates that our multifactorial approach offers improved accuracy (MAE < 8 years) in relation to other skeletal age estimation methods [137,139,140,141]. Independent validation of the method and software tools proposed here on samples from different temporal and biogeographic origins are of utmost importance to ascertain the broader impact and significance in archaeology, forensic anthropology, and medicine.

Artificial intelligence, statistical, and machine learning approaches are now ubiquitous in forensic and biological sciences. Several cases in the literature illustrate the usefulness of such approaches in adult macroscopic age-at-death estimation [13,14,15,22,24,124]. Although these approaches usually allow for flexible and non-parametric modeling with improved predictive performance, it also results in more opaque or black-box models from a non-expert perspective. These approaches also require proper validation and model selection techniques to avoid overfitting [142]. In this study, we applied a resampling approach to cross-validation based on Monte Carlo cross-validation for fair model assessment, and we also used a robust, analytical, computationally efficient leave-one-out cross-validation strategy to set the regularization parameter of the networks developed in experiments A and B. Randomization rather than optimization of the hidden layers, combined with an efficient C++ implementation of our models, allowed the construction of software that enables on-the-fly computation and validation (LOOCV) of deep architecture models for any combination of traits with minimal to no technical knowledge on the part of the user.

The problem of interpretability and explainability is a current issue in computational systems using machine learning techniques and constitutes an active topic of research in artificial intelligence [143]. A detailed methodological and implementation analysis will be the focus of a future work, but we briefly describe here how we handle the issue of explainability and interpretability in age-at-death using the neural networks with our software. As previously stated, we can look at the neural network fitted using the techniques described in this manuscript as a regularized linear model operating on the non-linear features extracted by the hidden layers concatenated with the original input (skip layer). We can exploit this property and use the intuitive and additive nature intrinsic to linear models and build a linear surrogate model to explain or interpret any neural network and its predictions.

In DRNNAGE, we regress the cross-validated predictions of the DRNN model on the original input of the network. We decorrelate the input data using the previously described sphering technique and standardize it to zero mean and unit variance. This results in a surrogate model where the intercept or baseline is the average of network estimates, and a new estimate can be “explained” by the sum of the contributions of individual traits to arrive at an approximation of the network estimate (Figure 8).

Our results suggest that a regression-based framework produces accurate age estimation in adult individuals. Prediction intervals can be estimated with ease and computational efficiency. Bayesian approaches [16,20,23] could have been used for this purpose but they encapsulate a different philosophy to data analysis and are more restrictive in regard to assumptions, parameterization, and computational efficiency compared to the ANN approach we pursued here. Recent contributions suggest that Bayesian approaches do not radically improve age-at-death estimation or outperform regression-based approaches [144,145].

The predictive modeling or function approximation approach pursued in this work is, at the same time, its strongest point and its key limitation. Although neural networks as function approximation machines allowed us to obtain individual accurate age estimates, a predictive modeling strategy—regardless of the underlying algorithm—can only demonstrate that there is an efficient mapping in the form of $y=f^{*}\left(x\right)$. Such a strategy does not explain the underlying biology of the skeletal traits. Fully understanding the biology of the skeletal traits used in age estimation is perhaps the greatest challenge of this problem, and perhaps the solution for more refined age estimation based solely on the skeletal morphology.

Despite the promising results, the current research did not emerge in a vacuum, nor has it any pretension to be a one-size-fits-all solution to skeletal age estimation, because it was inspired by significant work that was previously developed on this topic, see [16,19,24,35,140].

An important technical and methodological aspect that deserves a detailed analysis in the future is intra- and interobserver error. The results demonstrate the proposed scoring method is highly reproducible. This can be explained by the fact that most traits are encoded in a binary fashion; nonetheless, more data are required from an independent third party that applies the method as described here.

One last aspect that deserves discussion is the dataset employed in this study. The constructed dataset aimed to be uniform and homogeneous in respect to age-at-death and sex. At the moment, it only represents Portuguese nationals over a broad time span; thus, it would be important to expand the dataset to include individuals from other regions, and ascertain possible population and temporal differences in the performance of the proposed method.

## 5. Conclusions

The work presented here is an important and valuable contribution to the field of age-at-death estimation. Our results clearly demonstrated that a multifactorial approach improves accuracy and precision over single anatomic regions, as established in traditional adult skeletal aging methods. Multifactorial neural models introduce a two-to-six-fold reduction in the mean absolute error and prediction bias compared to standard models. This research also demonstrated that it is possible to produce informative age estimates for the elderly and that nonstandard skeletal traits are pivotal in the later stage of the adult age span. As an age estimation technique developed with forensic casework as its applicational domain, proper validation by other researchers and practitioners is most needed as we are aware that our results, as solid as they are, reflect only in silico performance and cross-validation. This work clearly demonstrated that neural network models offer excellent predictive accuracy. A current issue to be further investigated in future research work is the problem of interpretability and explainability. We briefly alluded to how this problem can be tackled using a global surrogate modeling approach, but other techniques will be investigated in the future so that age-at-death estimation can be approached with computationally accurate and intelligible techniques.

## Figures and Tables

**Figure 1 biology-11-00532-f001:**
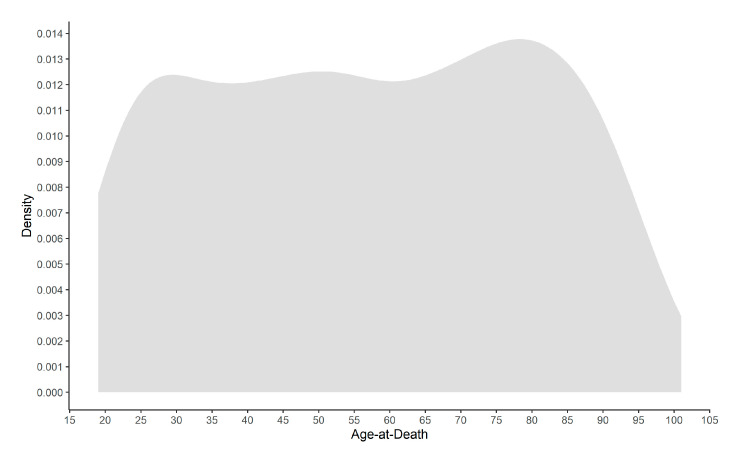
Pooled age-at-death distribution (KDE).

**Figure 2 biology-11-00532-f002:**
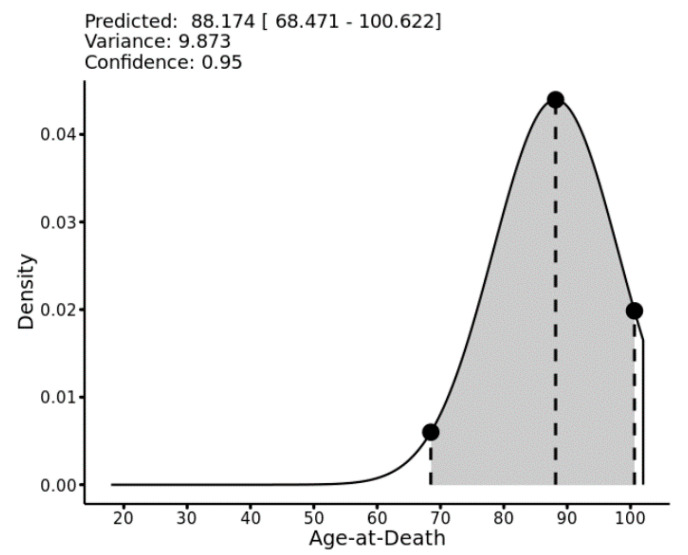
Prediction interval visualization using a (truncated) Gaussian uncertainty model.

**Figure 3 biology-11-00532-f003:**
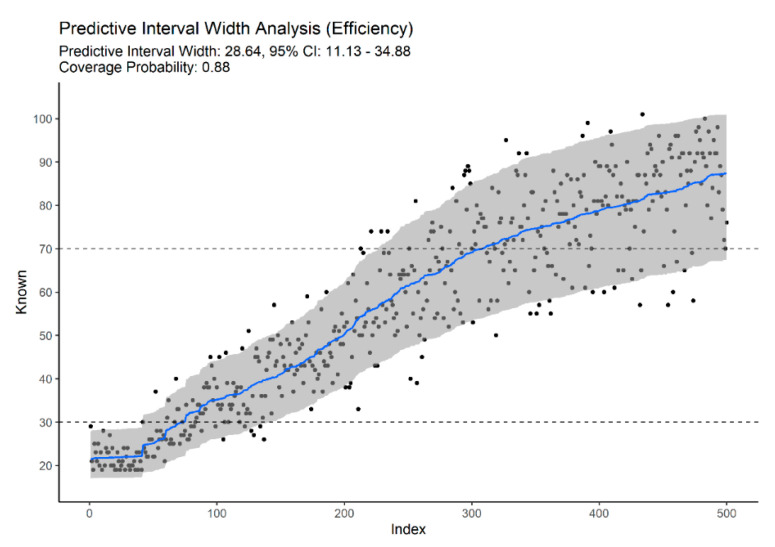
Predictive efficiency of standard age-related traits, α = 0.1.

**Figure 4 biology-11-00532-f004:**
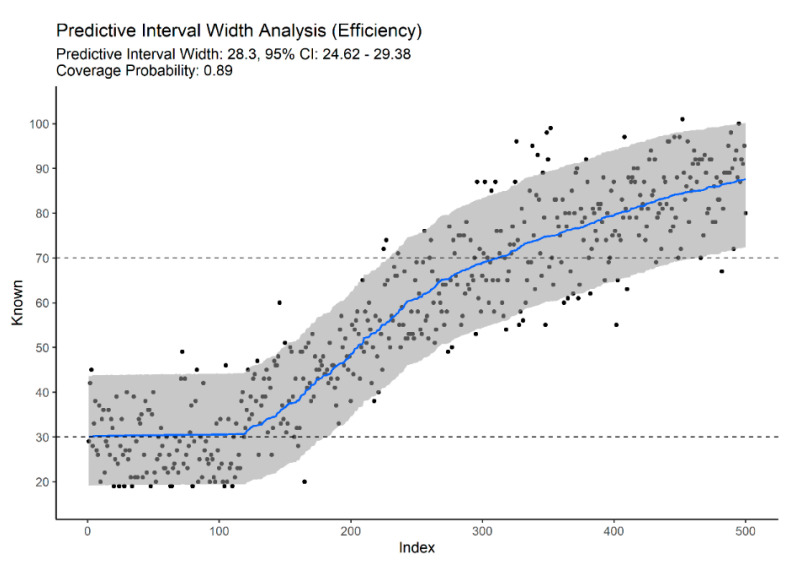
Predictive efficiency of degenerative traits of the axial and appendicular skeleton, α = 0.1.

**Figure 5 biology-11-00532-f005:**
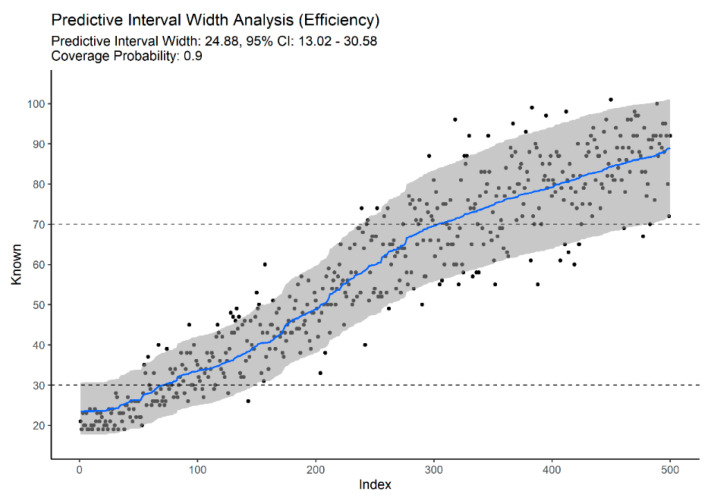
Predictive efficiency of full traits, DRNN-RUM model, α = 0.1.

**Figure 6 biology-11-00532-f006:**
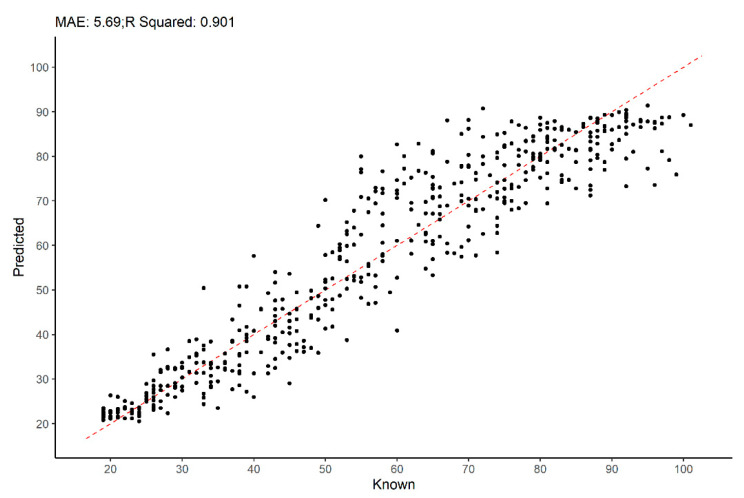
Known vs. predicted age-at-death using a full set of traits (LOOCV, *n* = 500).

**Figure 7 biology-11-00532-f007:**
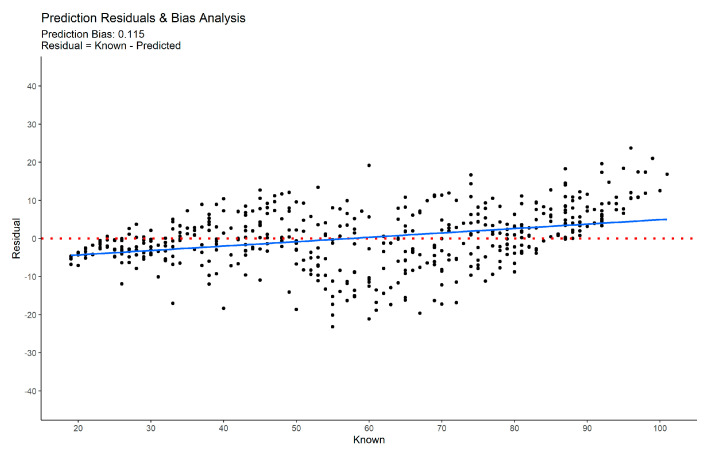
Prediction bias plot for the multifactorial (m = 64) RANN model.

**Figure 8 biology-11-00532-f008:**
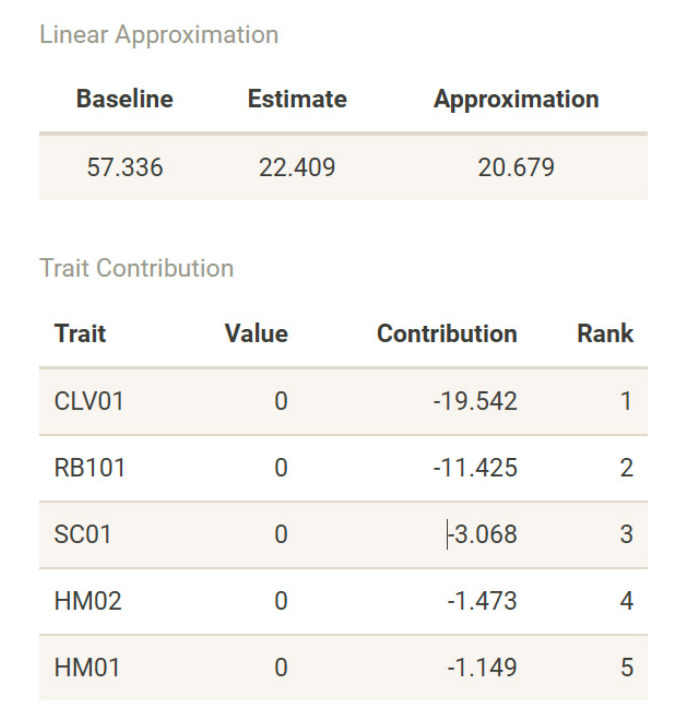
Explanation of an estimate by a linear surrogate model as performed by DRNNAGE software.

**Table 1 biology-11-00532-t001:** Demographic characterization of reference data sampled from the CISC and XXI-ISC collections.

		CISC		XXI-ISC		Pooled Collections		Pooled Sex
		Female	Male	Female	Male	Female	Male	
	*n*	168	166	82	84	250	250	500
Age-at-Death	Mean	48.482	45.331	81.841	74.881	59.424	55.260	57.34
(AGE)	Std. Dev.	19.483	18.171	12.889	15.082	23.556	22.141	22.93
	Min.	19	19	38	25	19	19	19
	Max.	95	96	101	96	101	96	101
Year of Birth	Mean	1877.286	1879.994	1923.866	1930.560	1892.564	1896.984	1894.774
(YOB)	Std. Dev.	21.252	19.948	13.137	14.424	28.969	30.096	29.591
	Min.	1830	1836	1904	1908	1830	1836	1830
	Max.	1911	1917	1970	1982	1970	1982	1982
Year of Death	Mean	1925.768	1925.325	2005.707	2005.440	1951.988	1952.244	1952.116
(YOD)	Std. Dev.	6.597	7.343	3.707	3.919	38.051	38.452	38.214
	Min.	1910	1910	2000	1995	1910	1910	1910
	Max.	1936	1936	2012	2011	2012	2011	2012

**Table 2 biology-11-00532-t002:** Monte Carlo cross-validation metrics for DRNN models built on pre-specified skeletal traits sets.

		Accuracy	Bias	Validity	Efficiency		
Traits		MAE	${\hat{\beta }}_{e}$	P(α)	PIW	PIW 95% CI	
Sutures	Median	15.300	0.656	0.950	68.144	51.699	69.759
(m = 9)	95% CI	13.586	0.590	0.900	66.054	46.361	68.312
		17.206	0.732	0.990	69.741	55.776	70.963
Axial	Median	8.185	0.198	0.960	38.754	33.732	40.842
(m = 16)	95% CI	7.365	0.137	0.920	37.102	32.272	39.215
		9.139	0.260	0.990	40.091	35.029	42.191
Appendicular	Median	7.583	0.167	0.960	37.378	29.109	39.541
(m = 23)	95% CI	6.678	0.103	0.910	35.412	27.613	38.014
		8.523	0.231	0.990	39.079	30.399	41.061
Clavicle	Median	8.949	0.244	0.960	49.234	17.354	51.610
(m = 2)	95% CI	7.798	0.169	0.920	39.064	15.981	49.962
		10.192	0.307	0.990	52.688	18.617	53.098
First Rib	Median	9.500	0.277	0.950	48.936	24.334	49.637
(m = 2)	95% CI	8.138	0.204	0.900	46.879	22.499	47.687
		10.831	0.351	0.990	50.903	26.078	51.533
Pubic symphysis	Median	10.897	0.370	0.940	51.210	26.905	56.954
(m = 3)	95% CI	9.371	0.280	0.870	48.688	24.520	54.799
		12.542	0.459	0.980	55.558	29.058	58.802
Sacroiliac complex	Median	8.523	0.223	0.950	44.668	20.378	47.969
(m = 6)	95% CI	7.380	0.145	0.890	39.350	18.596	46.017
		9.742	0.288	0.990	47.547	21.915	49.720
Acetabulum	Median	8.886	0.229	0.970	42.978	31.727	45.742
(m = 3)	95% CI	7.758	0.162	0.920	41.201	29.897	43.891
		10.006	0.287	1.000	44.509	33.240	47.304
Degenerative traits	Median	6.962	0.147	0.970	33.732	28.882	35.122
(m = 39)	95% CI	6.084	0.085	0.920	32.460	27.570	33.488
		7.814	0.200	1.000	34.935	30.019	36.656
Standard traits	Median	6.609	0.147	0.950	34.245	12.927	41.087
(m = 16)	95% CI	5.561	0.087	0.890	29.701	11.833	39.097
		7.598	0.202	0.990	37.857	14.169	42.833
All	Median	5.925	0.117	0.950	30.010	15.631	36.081
(m = 64)	95% CI	5.101	0.060	0.900	26.817	14.464	34.612
		6.728	0.170	0.990	33.191	16.811	37.515

**Table 3 biology-11-00532-t003:** Leave-one-out cross-validation metrics for DRNN models built on pre-specified skeletal traits sets.

		Accuracy	Bias	Validity	Efficiency		
Traits		MAE	${\hat{\beta }}_{e}$	P(α)	PIW	PIW 95% CI	
Sutures	Median	15.245	0.655	0.953	68.120	51.782	69.796
(m = 9)	95% CI	14.683	0.616	0.940	66.377	46.429	68.371
		15.751	0.692	0.963	69.708	55.878	70.996
Axial	Median	8.156	0.200	0.960	38.825	33.594	40.881
(m = 16)	95% CI	7.896	0.184	0.953	37.468	32.131	39.279
		8.394	0.213	0.968	39.872	34.902	42.234
Appendicular	Median	7.557	0.169	0.960	37.534	29.035	39.599
(m = 23)	95% CI	7.278	0.155	0.948	35.996	27.542	38.082
		7.823	0.184	0.970	38.920	30.319	41.109
Clavicle	Median	8.943	0.245	0.963	49.216	17.336	51.768
(m = 2)	95% CI	8.606	0.228	0.953	47.184	15.969	50.112
		9.248	0.263	0.970	51.238	18.597	53.252
First Rib	Median	9.409	0.275	0.950	48.897	24.356	49.811
(m = 2)	95% CI	9.067	0.255	0.938	47.036	22.502	47.862
		9.751	0.296	0.960	50.829	26.102	51.724
Pubic symphysis	Median	10.898	0.370	0.932	51.113	27.029	57.040
(m = 3)	95% CI	10.436	0.343	0.922	48.668	24.616	54.949
		11.315	0.398	0.945	53.003	29.217	58.909
Sacroiliac complex	Median	8.438	0.220	0.950	44.765	20.350	48.037
(m = 6)	95% CI	8.075	0.200	0.940	42.461	18.607	46.091
		8.741	0.239	0.960	46.755	21.893	49.800
Acetabulum	Median	8.833	0.229	0.965	43.051	31.541	45.832
(m = 3)	95% CI	8.490	0.210	0.955	41.302	29.726	43.995
		9.116	0.247	0.975	44.535	33.054	47.395
Degenerative traits	Median	6.929	0.147	0.963	33.744	28.816	35.194
(m = 39)	95% CI	6.694	0.133	0.953	32.530	27.499	33.566
		7.154	0.157	0.973	34.829	29.946	36.715
Standard traits	Median	6.561	0.145	0.948	34.283	12.952	41.170
(m = 16)	95% CI	6.277	0.132	0.935	32.464	11.853	39.222
		6.855	0.157	0.960	36.027	14.122	42.921
All	Median	5.899	0.118	0.950	30.057	15.558	36.141
(m = 64)	95% CI	5.677	0.110	0.940	28.758	14.403	34.644
		6.121	0.127	0.963	31.485	16.668	37.620

**Table 4 biology-11-00532-t004:** Monte Carlo cross-validation metrics for DRNN models built on different fractions of available skeletal traits.

		Accuracy	Bias	Validity	Efficiency		
Available Traits %		MAE	${\hat{\beta }}_{e}$	P(α)	PIW	PIW 95% CI	
90%	Median	5.964	0.120	0.950	30.354	15.851	36.215
(m ≈ 57)	95% CI	5.136	0.062	0.900	27.067	14.466	34.554
		6.773	0.169	0.990	33.422	18.081	37.705
80%	Median	6.026	0.121	0.950	30.498	16.004	36.261
(m ≈ 51)	95% CI	5.211	0.061	0.900	27.183	14.213	34.498
		6.851	0.172	0.990	33.584	18.492	37.902
70%	Median	6.072	0.125	0.950	30.805	16.206	36.454
(m ≈ 44)	95% CI	5.152	0.062	0.900	27.528	14.001	34.600
		6.924	0.180	0.990	34.004	19.666	38.405
60%	Median	6.131	0.125	0.950	30.964	16.352	36.649
(m ≈ 38)	95% CI	5.316	0.065	0.900	27.513	13.893	34.672
		7.049	0.179	0.990	34.320	20.532	38.692
50%	Median	6.237	0.129	0.950	31.479	16.717	36.969
(m ≈ 32)	95% CI	5.293	0.064	0.900	27.820	13.757	34.930
		7.180	0.179	0.990	34.854	22.119	39.250
40%	Median	6.360	0.134	0.950	32.125	17.165	37.429
(m ≈ 25)	95% CI	5.441	0.074	0.900	28.500	13.910	35.075
		7.380	0.193	0.990	35.636	23.292	40.166
30%	Median	6.570	0.140	0.950	33.163	17.933	38.137
(m ≈ 19)	95% CI	5.565	0.075	0.900	29.036	13.905	35.393
		7.651	0.201	0.990	36.916	25.407	40.861
20%	Median	6.951	0.153	0.950	35.263	19.946	39.694
(m ≈ 12)	95% CI	5.857	0.086	0.900	31.082	14.074	36.427
		8.139	0.218	0.990	39.625	28.892	43.619
10%	Median	8.026	0.196	0.950	39.618	26.914	43.025
(m ≈ 6)	95% CI	6.592	0.119	0.900	34.681	15.495	38.368
		9.683	0.276	0.990	46.043	34.276	49.479

**Table 5 biology-11-00532-t005:** Leave-one-out cross-validation metrics for DRNN models built on different fractions of available skeletal traits.

		Accuracy	Bias	Validity	Efficiency		
Available Traits %		MAE	${\hat{\beta }}_{e}$	P(α)	PIW	PIW 95% CI	
90%	Median	5.942	0.121	0.953	30.276	15.745	36.278
(m ≈ 57)	95% CI	5.699	0.110	0.940	28.748	14.339	34.599
		6.198	0.131	0.965	31.797	18.048	37.772
80%	Median	5.970	0.122	0.953	30.476	15.941	36.332
(m ≈ 51)	95% CI	5.702	0.108	0.940	28.860	14.162	34.574
		6.235	0.132	0.965	31.963	18.470	37.938
70%	Median	6.028	0.124	0.953	30.711	16.182	36.518
(m ≈ 44)	95% CI	5.737	0.108	0.938	28.960	14.013	34.697
		6.376	0.137	0.965	32.583	19.643	38.435
60%	Median	6.078	0.125	0.953	30.975	16.342	36.716
(m ≈ 38)	95% CI	5.768	0.108	0.938	29.070	13.872	34.756
		6.441	0.140	0.965	33.017	20.569	38.732
50%	Median	6.173	0.128	0.953	31.502	16.684	37.040
(m ≈ 32)	95% CI	5.819	0.111	0.938	29.410	13.724	34.989
		6.648	0.146	0.968	33.900	22.110	39.305
40%	Median	6.305	0.132	0.953	32.146	17.153	37.511
(m ≈ 25)	95% CI	5.903	0.114	0.935	29.839	13.905	35.130
		6.797	0.153	0.968	34.565	23.287	40.214
30%	Median	6.501	0.138	0.953	33.097	17.923	38.203
(m ≈ 19)	95% CI	6.046	0.118	0.935	30.583	13.899	35.468
		7.096	0.163	0.965	35.986	25.377	40.943
20%	Median	6.957	0.154	0.953	35.321	19.986	39.742
(m ≈ 12)	95% CI	6.316	0.127	0.935	32.096	14.117	36.479
		7.674	0.184	0.968	38.931	28.768	43.707
10%	Median	7.952	0.192	0.955	39.733	26.846	43.076
(m ≈ 6)	95% CI	6.968	0.154	0.940	35.229	15.515	38.419
		9.214	0.256	0.973	46.437	34.087	49.551

## Data Availability

Data used for this manuscript preparation is embedded in the software DRNNAGE and it is available from GitHub, see Section 2.6.4. Raw data is available upon request.

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
