# Peer review of "Adult Skeletal Age-at-Death Estimation through Deep Random Neural Networks: A New Method and Its Computational Analysis"

_biology, 2022, doi:10.3390/biology11040532_

Round 1

Reviewer 1 Report

This well-written and well-presented paper takes up the forensic problem of determining the age of an adult individual at death from their skeletal remains, using a neural networks method. The material consists of the skeletons (known age and sex) of individuals composing the reference collections kept at the University of Coimbra, 250 males and 250 females aged from 19 to 101 years, who died between 1910 and 2012. The analysis of 64 age-dependent morphological traits is done by an artificial neural network (ANN) method. Deep randomized network (DRNN) models were chosen. A new DRNNAGE software was developed and made available to users. The inter-observer error is minimal (average concordance coefficient = 0.907). The predictive value of this method with a mean absolute error of about 6 years is 2 to 6 times higher than that of conventional methods based on a single anatomical region. In order to test this method on practical, anthropological or forensic cases, it is important that this article be published making the software available for users. 

Author Response

Dear Reviewer 1,

We would like to thank you for the time and effort spent reviewing our manuscript,

Thank you for your review, we also believe that is important for our manuscript to be published so that the scientific community can access and test the proposed method and software,

Minor structure, language and spell-checking changes were performed in order to enhance the manuscript in accordance to your and other review reports,

Best regards,

The authors

Reviewer 2 Report

Dear Authors,

The article is too long, and readers get lost in its content. 
The keywords should be according to the international references (MeSH). They corrected it. 
Several sentences are not referenced (lines 49-55) and should be clarified. The introduction section looks more like a review of the authors' opinions than a literature review. 
The materials and methods section should be more curated and summarize the methods used. We kept with the idea that this is a literature review and its discussion.
The sentence "complete long bone development and rather than exclude an individual due to pathology or taphonomy on a specific skeletal trait, in that case, that trait was considered non-scoring and not recorded" (lines 153-155) expresses a criterion for inclusion of the samples. It is confusing for objective interpretation!
The results section does not identify the analytical tests selected in the intraobserver analysis. This section does not perform the interobserver analysis.
In the conclusions section, the authors do not present the conclusions of their work but rather their considerations as discussed sentences according to literature.

Author Response

Dear Reviewer 2,

Thank you for your time and effort reviewing our manuscript.

We will address your comments in a point-by-point fashion:

Point 1: "The article is too long, and readers get lost in its content."

Response 1: It is indeed a long manuscript. We can't stress enough how grateful we are for your review given that. We do believe that despite its length it is the appropriate way to present our work and that our work is indeed an important contribution to the topic. As we state it is methodological manuscript in nature and slicing it in two or more smaller manuscripts would devoid its true purpose: to propose a new multifactorial method for age estimation based on new scoring schemes optimized for that purpose and an advance computational approach based on deep random neural networks. We don't care about the parts; we care about the whole - age estimation from the skeleton as whole. Both the biological/anthropological and computational components are important, and they should be reflected in the manuscript corpus. This journal does not restrict manuscript length and discourages "salami publishing" strategies (see Instructions for Authors) nonetheless your comments encouraged us to improve certain parts of the manuscript to make it more comprehensive yet concise.

Point 2: "The materials and methods section should be more curated and summarize the methods used. We kept with the idea that this is a literature review and its discussion."

Response 2: We restructured and edited most of the text from section 2.2 to make it concise and straightforward. Lines 170 to 200 provide a rationale of our method development. The subsections of 2.2 provide background on existing scoring procedures and provide our proposed scoring system as a series of Tables and a brief discussion on the rationale behind method development and its difficulties. Lines 197 to 200 convey this idea in a clear manner to avoid the confusion of going through a "literature review and its discussion". Text on 2.2.3 was edited to make it more concise and devoid of unnecessary details. Regarding the computational methodology, it was left intact. We believe that it is important for the reader to understand how exactly ANN models were computed and how simple they are given the framework used. The lack of clarity regarding computation in age estimation is a pervasive problem. It is of utmost importance to understand mathematically how the estimation is obtained and evaluated.

Point 3: "The sentence "complete long bone development and rather than exclude an individual due to pathology or taphonomy on a specific skeletal trait, in that case, that trait was considered non-scoring and not recorded" (lines 153-155) expresses a criterion for inclusion of the samples. It is confusing for objective interpretation!"

Response 3: That sentence was indeed confusing, thank you. We restructured and edited section 2.1.1 (Lines 111-139) to make it more concise. That sentence was eliminated, and its idea is conveyed in Lines 128-129.

Point 4: "The results section does not identify the analytical tests selected in the intraobserver analysis. This section does not perform the interobserver analysis."

Response 4: The analytical test used to assess interobserver error was Kendall's W Concordance Coefficient. It was mentioned in section 2.2.9 Scoring Reliability: Intra-observer error. We now mention it in section 3.1 as well. No inter-observer was performed because the presented work results from the ongoing doctoral work by the first author. At the time of that data collection was not feasible to perform inter-observer error analysis. We mention in the manuscript that further analysis of inter- and intra-observer error by independent third parties is required (Lines 724-775, 1002-1006).

Point 5: "In the conclusions section, the authors do not present the conclusions of their work but rather their considerations as discussed sentences according to literature."

Response 5: The conclusions section was indeed on the weak and vague side, thank you. We improved it, clearly stating the main conclusions from our work (Lines 1014 - 1020).

Point 6: "The keywords should be according to the international references (MeSH). They corrected it."

Response 6: Dear reviewer we confess to be oblivious regarding the necessity of having keywords according to MeSH. To the best of our knowledge the only keyword used that is not MeSH compliant is "age-at-death estimation" the MeSH alternative is Age Determination by Skeleton which is inaccurate in the sense that one does not determine (implied high degree of certainty) but produce an estimate (implied uncertainty) of age from the skeleton.

Point 7: "Several sentences are not referenced (lines 49-55) and should be clarified. The introduction section looks more like a review of the authors' opinions than a literature review."

Response 7: We introduced minor edits to introduction deleting most of mentioned lines. They represented unnecessary and connective ideas not pivotal to the manuscript. We are not fully grasping your comment regarding the introduction. We introduce the modern state of forensic anthropology, identify biological profiling (age-at-death estimation) as a key attribution of the discipline despite its evolution. State how important is adult skeletal age-at-death estimation, identify the significant issues in this topic - mostly multifactorial age estimation and combination of different skeletal traits, say how we propose to deal with it in our research. highlight the major output of our project. Pretty standard. The most relevant sources are cited. Some of them review papers.

We hope that you find the changes introduced and responses to your comments satisfactory,

Once again thank you for your time and effort,

Best,

Reviewer 3 Report

The manuscript under review deals with the very important topic of adult skeletal age-at-death estimation, adopting a cutting edge computational approach, multiple skeletal markers, and a large and appropriate sample. Very importantly, the authors have designed an open-access user-friendly software that implements age-at-death estimation based on their proposed method. The paper makes a very important contribution to the field and I recommend its publication, though some revisions are requested first:

  1. I found the paper too long, which made it rather tiring to read, and I often felt I was missing important information due to the fact that too much redundant information was given. I would strongly urge the authors to considerably reduce its length.
  2. The authors correctly stress that their method needs to be validated in other assemblages. I believe that the fact that the collections used for its development are population-specific and time-specific, potentially limiting the applicability of this method in other contexts, should be stressed more in the text.
  3. There are several small syntax/grammar errors throughout the manuscript, thus a native speaker must edit it.
  4. How does this method deal with age mimicry effects?
  5. Table 1: The columns titled ‘Sex Pooled’ should actually be ‘Pooled Collections’, and the column titled ‘Pooled’ should be ‘Pooled sex’.
  6. The authors used data imputation to fill in missing data. They should report what percentage of the data was filled in using imputation. (for example, did their original dataset contain 30% missing data or 60% missing data?).
  7. As a general note, it would be helpful to add to the software website images visualizing different trait expression stages; however, this is certainly not necessary for the current publication purposes.
  8. Throughout page 25, Figures 2 to 4 are actually Figures 3 to 5.
  9. In Figures 3 to 5, it is not clear to me what α stands for. Is it the significance level? If yes, shouldn’t it be 0.05 since the CI is 95%. Also, is α the same as σ defined in line 915?

Author Response

Dear Reviewer #3,

Thank you for your time and effort reviewing our manuscript.

We will address your comments in a point-by-point fashion:

Point 1: "I found the paper too long, which made it rather tiring to read, and I often felt I was missing important information due to the fact that too much redundant information was given. I would strongly urge the authors to considerably reduce its length."

Response 1: It is indeed a long manuscript. We can't stress enough how grateful we are for your review given that. We do believe that despite its length it is the appropriate way to present our work and that our work is indeed an important contribution to the topic. As we state it is methodological manuscript in nature and slicing it in two or more smaller manuscripts would devoid its true purpose: to propose a new multifactorial method for age estimation based on new scoring schemes optimized for that purpose and a computational approach based on deep random neural networks. We don't care about the parts alone; we care about the whole - age estimation from the skeleton as whole. Both the biological/anthropological and computational components are important, and they should be reflected in the manuscript corpus. This journal does not restrict manuscript length and discourages "salami publishing" strategies (see Instructions for Authors) nonetheless your comments encouraged us to improve certain parts of the manuscript to make it more comprehensive yet concise. We were not able to reduce it in size per se, but we restructured and edited some parts of the text where the redundancy you mentioned was more visible, breaking the flow of the text. We restructured and edited most of the text from section 2.2 to make it concise and straightforward. Lines 170 to 200 provide a rationale of our method development. The subsections of 2.2 provide background on existing scoring procedures and provide our proposed scoring system as a series of Tables and a brief discussion on the rationale behind method development and its difficulties. Lines 197 to 200 convey this idea in a clear manner. Text on 2.2.3 was edited to make it more concise and devoid of unnecessary details. Regarding the computational methodology, it was left intact. We believe that it is important for the reader to understand how exactly ANN models were computed and how simple they are given the framework used. The lack of clarity regarding computation in age estimation is a pervasive problem. It is of utmost importance to understand mathematically how the estimation is obtained and evaluated. We introduced minor edits to the introduction deleting unnecessary and connective text not pivotal to the manuscript.

Point 2: "The authors correctly stress that their method needs to be validated in other assemblages. I believe that the fact that the collections used for its development are population-specific and time-specific, potentially limiting the applicability of this method in other contexts, should be stressed more in the text."

Response 2: We introduced lines 1002-1006 in the Discussion - "One last aspect that deserves discussion is the dataset employed in this study. The constructed dataset aimed to be uniform and homogeneous in respect to age-at-death and sex. At the moment it only represents Portuguese nationals over a broad time span, it would be important to expand the dataset with individuals from other regions and ascertain possible population and temporal differences in the performance of the proposed method."

Point 3: There are several small syntax/grammar errors throughout the manuscript, thus a native speaker must edit it.

Response 3: The text extensively edited.

Point 4: "How does this method deal with age mimicry effects?"

Response 4: We believe that no strategy can totally cope with the so-called age-mimicry either from a computational point standpoint or from a sampling standpoint. Nonetheless we opt for a sampling strategy by forming a homogeneous and uniform age-at-death distribution in our reference data. "An homogenous and uniform age-at-death distribution is a simple yet vital strategy to cope with the problem of age-mimicry [45] and to guarantee that targeted age span is fully represented in first place." (Lines 133-135). Some degree of age-mimicry is always present because the true distribution from which the deceased originated is unknown. There is always an implicit assumption that reference data represents such population or more broadly speaking it can capture the relationship between the skeletal changes and age-at-death

Point 5: "Table 1: The columns titled ‘Sex Pooled’ should actually be ‘Pooled Collections’, and the column titled ‘Pooled’ should be ‘Pooled sex’."

Response 5: Corrected as suggested.

Point 6: "The authors used data imputation to fill in missing data. They should report what percentage of the data was filled in using imputation. (for example, did their original dataset contain 30% missing data or 60% missing data?)."

Response 6: Lines 163-165 added - "Missing values represented 9.52% of the total entries of the data table when bilateral data is considered and 6.89% when the domain heuristic described was first applied as a naïve imputation mechanism and strategy to handle bilateral data redundancy."

Point 7: "As a general note, it would be helpful to add to the software website images visualizing different trait expression stages; however, this is certainly not necessary for the current publication purposes."

Response 7: It will be addressed in later stages of development and as the method is tested by independent researchers. The current goal is to have a fully functional and open-source software so that everyone can audit it and contribute to it.

Point 8: "Throughout page 25, Figures 2 to 4 are actually Figures 3 to 5."

Response 8: Thank you for pointing that out. Corrected.

Point 9: "In Figures 3 to 5, it is not clear to me what α stands for. Is it the significance level? If yes, shouldn’t it be 0.05 since the CI is 95%. Also, is α the same as σ defined in line 915?

Response 9: α (alpha) stands for the uncertainty level used on computed the Predictive Intervals (PI) for illustrative purposes. the 95% Confidence Intervals (CI) are in respect to the predictive interval width distribution.

We hope that you find the changes introduced and responses to your comments satisfactory,

Once again thank you for your time and effort,

Best,

Round 2

Reviewer 2 Report

Dear Authors,
Content selection is a scientific skill. The authors have done extensive work. Meanwhile, the manuscript should include the necessary and appropriate content for the reader in each section. If the authors' interest is to present an exhaustive text, perhaps this type of publication (periodical) is not the most appropriate. 
In response to point 4) the authors state that this is a continuing doctoral work of the first author. They assumed that the same individual collected the data. It should be emphasized the interobserver study is necessary for the methodological analysis, so I consider that its absence is a severe flaw. The authors should rethink the presentation of the methodology and be more restrained in their conclusions regarding their statement.
How can authors increase the impact of this publication if they do not promote its proper dissemination in the scientific community by MeSH terms?
In the introduction section, several sentences are referenced. The discussion sentences, authored by the authors, should be considered in the discussion section.

Author Response

Dear Reviewer #2

Once more thank you for your time and effort reviewing our manuscript.

As previously done, we will try to address your comments in a point-by-point fashion:

Point 1: "(...) Content selection is a scientific skill. The authors have done extensive work. Meanwhile, the manuscript should include the necessary and appropriate content for the reader in each section. If the authors' interest is to present an exhaustive text, perhaps this type of publication (periodical) is not the most appropriate. (...) The authors should rethink the presentation of the methodology (...)"

Response 1: As previously argued and stated, this is indeed a lengthy manuscript, but we do believe that despite its length it is the appropriate way to present our work and that our work is indeed an important contribution to the topic. As we stated it is methodological manuscript in nature and slicing it in two or more smaller manuscripts would devoid its true purpose. The work here presented unifies more than 30 years of research in macroscopic adult skeletal age estimation in single method and software. Going through the manuscript multiple times we have recognized that the presentation of the methodology can indeed be improved. We move the fifteen tables that compose our proposed scoring system to the Supplementary Material. We believe that in addition to the previous revisions to this section of the methodology it improves reading flow and overall manuscript presentation without sacrificing our vision. We sustain our point of view regarding the computational component of the methodology. We believe that it is important for the reader to understand how exactly ANN models were computed and how simple they are given the framework used. The lack of clarity regarding computation in age estimation is a pervasive problem. It is of utmost importance to understand mathematically how the estimation is obtained and evaluated.

Point 2: "In response to point 4) the authors state that this is a continuing doctoral work of the first author. They assumed that the same individual collected the data. It should be emphasized the interobserver study is necessary for the methodological analysis, so I consider that its absence is a severe flaw."

Response 2: This issue was previously addressed. What we meant to say was that the work presented was conducted in the context of the doctoral studies of the first author who collected all the data and performed the intra-observer error analysis. That is now clearly stated in the manuscript. We previously stated that at the time of that data collection was not feasible to perform inter-observer error analysis. It would be a severe flaw if no observer error analysis were performed or if the problem were not acknowledged and recognized at all. This limitation of our study was previously highlighted in the discussion - "An important technical and methodological aspect that deserves a detailed analysis in the future is intra- and interobserver error. The results demonstrate the proposed scoring method is highly reproducible. This can be explained by the fact that most traits are encoded in a binary fashion, nonetheless more data is required from an independent third party that applies the method as described here."

Point 3: "How can authors increase the impact of this publication if they do not promote its proper dissemination in the scientific community by MeSH terms?"

Response 3: This point was previously addressed. The MeSH (Medical Subject Headings) is the National Library of Medicine controlled vocabulary thesaurus used for indexing articles for PubMed. There is more to proper dissemination than PubMed indexing. We previously stated that we were not aware about the necessity of having keywords according to MeSH. To the best of our knowledge the only keyword used that is not MeSH compliant is "age-at-death estimation" the MeSH alternative is Age Determination by Skeleton which is inaccurate in the sense that one does not determine (implied high degree of certainty) but produce an estimate (implied uncertainty) of age from the skeleton. There are several publications indexed in PubMed using that exact keyword

Point 4: "(...) The authors should (...) be more restrained in their conclusions regarding their statement (...)."

Response 4: Our conclusions reflect the results obtained. We highlight the advantages and limitations of the work presented here and suggest aspects to improve/investigate in the future.

Point 5: "In the introduction section, several sentences are referenced. The discussion sentences, authored by the authors, should be considered in the discussion section."

Response 5 This point was previously addressed. In response to Point 7 of the previous review we stated "we introduced minor edits to introduction deleting most of mentioned lines. They represented unnecessary and connective ideas not pivotal to the manuscript. We are not fully grasping your comment regarding the introduction. We introduce the modern state of forensic anthropology, identify biological profiling (age-at-death estimation) as a key attribution of the discipline despite its evolution. State how important is adult skeletal age-at-death estimation, identify the significant issues in this topic - mostly multifactorial age estimation and combination of different skeletal traits, say how we propose to deal with it in our research. highlight the major output of our project. Pretty standard. The most relevant sources are cited. Some of them review papers".

We hope that you find the changes introduced and responses to your comments satisfactory,

Once again thank you for your time and effort,

Best,

This manuscript is a resubmission of an earlier submission. The following is a list of the peer review reports and author responses from that submission.